# From Forgotten Pathogen to Target for New Vaccines: What Clinicians Need to Know about Respiratory Syncytial Virus Infection in Older Adults

**DOI:** 10.3390/v16040531

**Published:** 2024-03-29

**Authors:** Matteo Boattini, André Almeida, Sara Comini, Gabriele Bianco, Rossana Cavallo, Cristina Costa

**Affiliations:** 1Microbiology and Virology Unit, University Hospital Città della Salute e della Scienza di Torino, 10126 Turin, Italy; gabrielebnc87@gmail.com (G.B.);; 2Department of Public Health and Paediatrics, University of Torino, 10126 Turin, Italy; 3Lisbon Academic Medical Centre, 1649-028 Lisbon, Portugal; 4Department of Internal Medicine 4, Centro Hospitalar Universitário de Lisboa Central, Centro Clínico Académico de Lisboa, 1169-024 Lisbon, Portugal; andre.almeida@chlc.min-saude.pt; 5NOVA Medical School, Universidade Nova de Lisboa, Centro Clínico Académico de Lisboa, 1169-056 Lisbon, Portugal; 6Operative Unit of Clinical Pathology, Carlo Urbani Hospital, 60035 Jesi, Italy; 7Department of Experimental Medicine, University of Salento, Via Provinciale Monteroni n. 165, 73100 Lecce, Italy

**Keywords:** respiratory syncytial virus, RSV, older adults, elderly, vaccine, pneumonia

## Abstract

Respiratory syncytial virus (RSV) is increasingly recognized as being implicated in acute illness in older adults, with a significant weight in hospitalizations for respiratory illness and death. By means of a best-evidence review, this paper aims to investigate whether RSV can be considered a forgotten pathogen in older patients, looking at trends in the literature volume and exploring possible epidemiological and clinical features underlying the focus given to it. We then present an assessment of its disease burden and present and future strategies for its reduction, particularly in light of the recent availability of new vaccines.

## 1. Introduction

Respiratory syncytial virus (RSV) is a well-recognized cause of acute respiratory tract illness among infants, with the first case of RSV-related bronchiolitis described as early as 1957 [1]. However, RSV is also increasingly recognized as being implicated in acute illness in older adults [2,3,4,5,6,7,8,9,10,11,12,13,14,15], contributing to a considerable burden in hospitalizations for respiratory illness while presenting mortality rates that are close to those associated with illness caused by influenza viruses [16,17,18]. 

RSV is transmitted through contact between viral particles from infected individuals and oral, nasal, and conjunctival mucous membranes. This contact can happen through the emission of droplets by direct contact or by self-inoculation after touching contaminated surfaces, where the virions can remain viable for several hours [19]. Direct contact is, however, the most common route of transmission [20,21,22]. RSV typically causes seasonal epidemics worldwide. In temperate climates, these usually occur in winter, whereas in tropical and semitropical climates, seasonal epidemics are usually associated with rainy seasons, and in some contexts, RSV circulation may be documented in as many as eight months of the year [23]. However, these seasonal variations have been disrupted by the COVID-19 pandemics and its containment measures [24]. At present, it is therefore not possible to state with absolute certainty that the circulation of RSV will resume in a manner similar to the pre-pandemic period [25]. 

In adults, the severity of the acute RSV illness is exacerbated by the presence of respiratory and cardiac comorbidities [10,13,26,27,28,29,30,31,32,33,34,35,36,37,38] as well as immunosuppression associated with bone marrow or solid organ transplantation [30,39]. Evidence shows that, after the acute respiratory illness resolves, the virus interferes with the immune system’s ability to establish memory, allowing for recurrent infections, particularly reinfections in the same winter season [40]. Therefore, RSV infection incidence is heterogeneous given RSV infection results in an incomplete natural immunity that predisposes hosts to reinfection and limits the implementation of serological studies to define the disease burden [23]. In the last decade, a growing body of literature has begun to point out that RSV is the second pathogen after *Streptococcus pneumoniae* more often implicated globally in deaths of people of all ages with a lower respiratory tract infection as well as causing a burden of disease comparable to that of influenza among people over 70 years of age [41]. The increased clinical focus has been accompanied by considerable interest and efforts in the development of new therapies against RSV, with two vaccines recently approved for active immunization of people over 60 years of age [42,43]. 

## 2. Research Questions

In this work, we aimed to provide evidence and comment on the following questions: (a)Is RSV a forgotten pathogen among older adults? Is there a shift in trend in the importance that is being given to it?(b)Does RSV disease among older adults present a considerable burden?(c)What new and developing strategies are there to reduce the need for medical care in older adults? Are these relevant?

## 3. Materials and Methods

We conducted a best-evidence review by searching all peer-reviewed journals focusing on RSV lower respiratory tract infections among older adults, emphasizing articles published since 2000. PubMed/MEDLINE and Cochrane database were scrutinized for articles using the following key word combinations: “respiratory syncytial virus” and “adults”, “respiratory syncytial virus” and “elderly”, “respiratory syncytial virus” and “older adults”, “respiratory syncytial virus” and “vaccine”. After providing an overview on RSV’s microbiology and epidemiology, we compared the RSV and influenza virus publication volumes in order to make a rough assessment of the clinical interest in the respective topics. We included clinical studies, reports, reviews, and meta-analyses. We excluded protocols, case reports, and case series as well as studies that were not specific for patients over 60 years of age and those that did not have RSV lower respiratory tract infections as its main subject. We then retraced the course of the literature on RSV in older adults. We identified the evidence in support of RSV as a forgotten pathogen, outlining its aspects and inconsistencies from a clinical, diagnostic, and surveillance point of view. Finally, we illustrated the pipeline of vaccines and new therapeutics. 

## 4. RSV Microbiological and Epidemiological Overview

RSV is an enveloped RNA virus and a member of the Paramyxoviridae family and Pneumovirinae subfamily. Virions are characterized by dimensions of 150 nm or more, usually spherical and made of negative single-stranded ribonucleic acid (3′–5′). This molecular characteristic means that in order to be infectious, it needs to be accompanied by polymerase to generate a chain capable of being translated into protein elements, which is why it is essential for the nucleocapsid to be internalized when it enters the host cell [23,44]. RSV presents minimal antigenic heterogeneity with nucleocapsid proteins (N, P, L e M2-1), envelope and non-structural proteins (M, NS1 e NS2), and transmembrane proteins (F, G e SH) [45]. The role of the G protein in attachment to the host cell and the role of the F protein in penetration into the host cell, as well as fusion between viral and host cell membranes, should be emphasized. This ability is responsible for the formation of syncytia, multi-nucleated cell clusters. Both the G and F proteins are important antigenic targets for immune neutralization. Variability in the F protein correlates with disease severity and effectiveness of monoclonal drug and vaccine development [46]. There are two subtypes of RSV described, A and B, with significant genomic variation between the two that are simultaneously present in most outbreaks, A subtypes typically causing the more severe disease. The phenotypic differences are expressed above all in terms of the antigenic expression of the G protein [47].

As far as older adults are concerned, data from US health systems and epidemiological surveillance suggest that RSV is responsible for up to 10,000 deaths per year in people over 64 as well as 60,000 to 120,000 hospitalizations in this population [48]. Data from the city of London from 1994 to 2011 suggest an incidence of 0.7 hospitalizations per 1000 inhabitants [49]; in New York from 2017 to 2020, the data suggest an incidence of 0.4 to 0.6 per 1000 [5]. The only study available from a tropical context, conducted in Guatemala and based on surveillance data, suggests an incidence of 0.3 per 1000 per year [50]. 

## 5. Literature Overview Output

The volume of publications on RSV or influenza viruses in older adults over the last 23 years are reported in Figure 1. 

As a main paper topic, RSV among adults/elderly/older adults was given 7.9 (RSV OR influenza AND elderly) to 11.2 (RSV OR influenza AND vaccine) times less publication volume than its counterparts in influenza. At the same time, it was noted that attention to influenza has always been relatively constant while that for RSV has been on the rise. We finally found a total of 1452 articles that matched our key words, from which we ultimately included a total of 86 for the purposes of this review.

## 6. Can We Consider RSV a Forgotten Pathogen in Older Adults?

Although “forgotten pathogen” is a formulation that is found every now and then in the scientific literature, especially when referring to infrequent diseases, it does not pertain to a rigorous standardized definition. A search for the term on PubMed/MEDLINE yields just 14 results, with the main suspects being *Streptococcus pneumoniae*- and *Chlamydophila psittaci*-related infections and no results for RSV. Similarly, a search for the term “forgotten disease” provided 287 results with the majority of published papers referring to cases of Lemierre’s syndrome. Therefore, these terms seem to be used as narrative artifices to draw the reader’s attention rather than to define a clinical underestimation of a given infectious agent in a given population. Although it does not outline the object, the absence of a definition of forgotten pathogen draws attention to the actors involved in forgetting. In the case of RSV in older adults, clinicians, lab diagnostics, surveillance system, and drug companies could have been involved in this process.

### 6.1. Literature Focus of RSV Infection in Older Adults

The scientific literature focus on RSV in older adults is quite recent. Falsey et al. were among the first to take an interest in the topic with some of their works on factors associated with disease severity and mortality remaining the most common references in the field [2,3]. In the last four years, several systematic reviews and meta-analyses were published [15,16,17,25,27,29,31,32,41], most of them with the aim to define the disease burden of RSV in older adults and some of them with the active participation of pharma companies involved in RSV vaccine and drug development. Several clinical studies have been published over the years worldwide [51,52,53,54,55,56,57,58,59,60,61,62,63,64,65,66,67,68,69,70,71,72,73,74,75,76,77,78,79,80,81]. In Europe, what emerged together with the findings of few multicenter clinical studies [21,59,60,61,62,63,64,65] was the need to harmonize and collaborate between the various laboratories in order to obtain shared diagnostic protocols with which to compare data [82].

### 6.2. Factors Supporting RSV Being a Forgotten Pathogen in Older Adults

Several factors have been reported to be associated with underestimating the RSV disease burden in patients over 60 years [31,83]. These factors relate mainly to clinical features (e.g., non-specificity of symptoms, lack of an effective treatment, absence of a licensed vaccine), diagnostics aspects (e.g., low sensitivity of the antigen-based tests, high costs of polymerase chain reaction (PCR) diagnostic techniques, and presence of low viral loads in older adults), and surveillance-related aspects (e.g., lack of a clinical case definition, non-specificity of the surveillance system).

#### 6.2.1. Clinical Picture and Disease Management

After replicating in the nasopharynx, RSV infects the small bronchiolar epithelium, then extends to alveolar pneumocytes. Pathologic findings of RSV include the necrosis of epithelial cells, occasional proliferation of the bronchiolar epithelium, and infiltration of inflammatory cells between the vascular structures and small airways. This leads to airway obstruction, air trapping, and increased airway resistance [23,84]. However, the symptomatology is non-specific, with an average incubation time of 4–6 days. An asymptomatic infection, a mild infection with symptoms of an upper respiratory tract infection, and a more severe infection with symptoms of a lower respiratory tract infection that can also lead to decompensation of chronic pulmonary and cardiac diseases can be considered the main spectrum of RSV infection presentation (Figure 2) [7,85,86].

Radiology also does not provide much guidance for a diagnosis, as chest tomography findings are very non-specific, as shown by a recent meta-analysis in which the authors, however, state that healthy adults with an RSV infection would have an increased risk for septal thickening, nodular lesions, and ground glass opacities compared to children [87].

Another clinical element that may have contributed to RSV’s oblivion in older adults is that there is no effective therapy other than symptomatic, supportive, and complication treatment. Nebulized or oral ribavirin has been shown to reduce morbidity and mortality in hematopoietic stem cell transplant recipients [39,88] but has not been as effective in the immunocompromised and older adults [30,89]. Treatment with glucocorticoids, albeit widely used, is controversial, there being outdated studies in the literature, such as this one showing that short courses of steroids did not affect viral load or shedding [90] and has been associated with longer hospital stays and secondary infections [11], thus leaving room for new studies in this area.

As multicenter clinical studies on RSV infections in older adults, the scenario is quite fragmented, with mortality rates ranging from 0 to 25.9% (Table 1) [21,51,52,53,54,55,56,57,58,59,60,61,62,63,64,65,66,67,68,69,70,71,72,73,74,75,76,77,78,79,80,81]. 

Most clinical studies have focused on investigating factors associated with the development of a low respiratory tract infection, especially at the population level, the risk of hospitalization, ICU admission or the need for ventilatory support, and mortality, often compared to a control population (especially patients with influenza virus infections) (Table 2). 

In Europe, a Portuguese, Italian, and Cypriot project that included patients hospitalized due to a lower respiratory tract infection due to influenza and/or RSV during two consecutive winter periods showed that influenza A H1N1 virus was the only microbiological factor associated with in-hospital mortality and a need for invasive mechanical ventilation [96]. In one of the sub-analyses carried out on the population over 85 years of age, however, it emerges that chronic diseases, especially COPD or asthma and chronic kidney disease KDIGO stage 3A or worse, weigh more on outcomes (pneumonia and death) than viral etiology [97]. In another sub-analysis of the same cohort, which included 166 elderly patients with a lower respiratory tract RSV infection, about 30% of the patients had pneumonia, the in-hospital mortality rate was 12.1%, and the factors associated with mortality were the male gender, solid neoplasia and obstructive sleep apnea, and/or obesity hypoventilation syndrome [62]. In a French multicenter study that included more than 1000 adult patients, the overall mortality rate was lower (around 6%), while it was more similar to the overmentioned cohort for ICU patients. The study reported the factors associated with a need for mechanical ventilation to be chronic cardio-pulmonary diseases [64]. In a Scottish and Danish study, Osei-Yeboah et al. found an increased risk of RSV hospitalization in adults with chronic obstructive pulmonary disease, ischemic heart disease, and chronic kidney disease when compared with those of the overall population [95].

#### 6.2.2. Diagnostic Challenges

Laboratory diagnostic methods for an RSV diagnosis mainly include rapid antigen testing and PCR. Despite their ease of use, antigenic tests for RSV do not have high sensitivity in adult patients [98,99]. Besides not being cheap and gold standard, PCR techniques have different sensitivities, with single-target PCRs having a higher sensitivity than multiplexes [100]. In a recent meta-analysis, the authors conclude that, for improving RSV detection, at least one other respiratory sample should be tested in addition to the RT-PCR test of the nasopharyngeal swab and that diagnostic sensitivity could benefit from testing up to at least 3 respiratory samples [101], increasing the complexity in terms of logistics and costs.

#### 6.2.3. Recent Developments on Surveillance and Case Definition

RSV surveillance was officially introduced by the WHO in 2016 [101], starting to take into account that, in about 50% of older adults, an RSV infection may occur without fever. A case definition for hospital and community was provided [102]. Extended SARI (severe acute respiratory infection) and ARI (acute respiratory infection) were defined, and the case definitions of influenza-like illness and SARI for influenza [103], which required fever as a symptom and on which both RSV surveillance and clinical studies had previously been based, were abandoned. The RSV surveillance system now includes screening for RSV of patients with symptoms compatible with extended SARI in sentinel hospitals [104]. For older adults, the respiratory sample to be tested is sputum. If the test is positive for RSV, typing in A or B and subsequent notification is performed. If the test is negative and the patient presented with fever, testing for influenza is performed, with the subsequent notification of the result [105].

## 7. Estimating RSV Disease Burden in Older Adults

In recent years, several meta-analyses aimed at estimating the RSV disease burden in adults have been published, most of them highlighting the high burden of the disease in older adults [16,27,28,29,31,32]. The first estimate was by Falsey et al., which stated that global incidence could be around 6.3 per 1000 people older than 70 years of age or high-risk adults [2]. A more recent meta-analysis estimated that, in 2015, RSV would cause illness in about 1.5 billion people over 50 years of age, that 14.5% would be hospitalized, and that 1.6% would die [28]. Another meta-analysis estimated a case fatality proportion between 8 and 10 percent in elderly and high-risk adult patients, respectively [27]. In Europe, a Northern European, prospective, observational cohort study aiming at assessing the community burden of RSV in older adults aged ≥ 60 years showed that RSV had an incidence between 4 to 7.2%, with mostly mild infections, leading to both fewer medical appointments and antibiotic prescriptions in comparison to influenza-associated infections [35]. Conversely, an estimation of hospital admission in Europe showed that RSV causes about 160,000 admissions per year, with about 92% of cases occurring in adults aged ≥ 65 years [106]. 

### 7.1. Limitations in Estimating RSV Disease Burden in Older Adults

Almost all the works that attempted to estimate the RSV disease burden in older adults reported several limitations. These are basically due to the processing data from studies performed with very different methods and protocols [107]; being based on the case definition of influenza-like illness [108]; lacking the necessary data to stratify by age group; being based on data mainly derived from academic, high-income, in-hospital settings [31,67]; and underestimating the presence of other factors, such as coinfections or complications [15]. To address these limitations, the WHO published a document with the essential guidelines to be followed in estimating the RSV disease burden [109], but robust evidence is lacking.

### 7.2. Burden of Coinfections

Among older adults with an RSV infection, the presence of another pathogen in the respiratory tract is another factor to be considered in the disease burden analysis [66]. The rates of viral and bacterial coinfection in causing pneumonia range from 10 to 68% [110], making the associated disease severity a wide area needing investigation. Several bacterial and viral respiratory pathogens, such as *Staphylococcus aureus*, *Pseudomonas aeruginosa*, *Streptococcus pneumoniae*, *Haemophilus influenzae*, *Moraxella catarrhalis*, *Klebsiella pneumoniae*, human rhinoviruses, influenza A virus, human metapneumovirus, and human parainfluenza viruses, have been reported to be involved [91,92,94,111]. Celante et al. reported a coinfection rate of 18.2% in their cohort of French hospitalized adults with an RSV infection, and coinfection was associated with a need for invasive mechanical ventilation [64]. Godefroy et al. showed that bacterial coinfections may be present in about 12% of older adults with an RSV infection, and these patients showed a higher mortality rate [93].

## 8. Pipeline of Vaccines and New Therapeutics

RSV vaccine candidates for pediatric, maternal, or older populations use four approaches: live attenuated, protein based (inactivated, subunit, or particle-based), nucleic acid vaccines, and recombinant vectors [112]. There are several candidates in different stages of study and development [113]. As far as the elderly are concerned, in spring 2023, two vaccines were approved by the responsible drug agencies and placed on the market for the active immunization of people over 60 years of age. Specifically, the first one, marketed by GlaxoSmithKline, is a monovalent vaccine and comprises RSV pre-F protein. It showed a vaccine efficacy of 94.1% in phase 3 trial (NCT04886596) against severe RSV-related lower respiratory tract disease and a satisfactory safety profile [114]. The second one, marketed by Pfizer, is a bivalent vaccine comprising subtype A and B of RSV pre-F proteins. The RENOIR phase 3 trial (NCT05035212) showed a vaccine efficacy of preventing an RSV-associated lower respiratory tract illness between 66.7 (at least two signs or symptoms) and 85.7% (at least three signs or symptoms) [115] and no vaccine-related serious adverse events through 12 months post-vaccination [116]. Concerning RSV vaccine candidates in Phase 3, the Moderna candidate, the only nucleic acid vaccine in phase 3, showed 83.7% (95.88% CI, 66.0–92.2) efficacy (NCT05127434) in preventing confirmed lower respiratory tract RSV infections with at least two lower respiratory symptoms with no safety concerns [117]. Of note, there are several phase 1 studies ongoing, assessing the safety and reactogenicity of combined mRNA vaccines in healthy older adults [118]. As far as new therapies are concerned, monoclonal antibodies that seem to be effective in newborns need to be assessed for efficacy and safety in at-risk adults, currently having no role. There are still other drugs that attempt to target the mechanism of virus entry and replication, some of which have shown promising results, such as fusion [119] and nucleoprotein inhibitors [120].

## 9. Conclusions

Although the term ‘forgotten pathogen’ is more akin to a narrative gimmick than a scientific term, one can fairly say that publication interest in RSV among older adults has been relatively low for a long time up until the late 2010s. Several reasons associated with the clinical characteristics of the associated infections, diagnostics, and surveillance have been put forward as contributing to this relative lack of concern. The surveillance system itself is very recent, compared to the influenza system, which is over 60 years old, so a full understanding of the phenomenon is still some time away. Moreover, the limitations of published studies make it difficult to estimate the real disease burden of this pathogen in the older and adult population in general. Interest in RSV among older adults has, however, grown in recent years, in parallel with the final stages of vaccine development, even though the associated disease burden has not been well characterized. In fact, in spite of this knowledge gap, it can be said that RSV has not been forgotten by the companies involved in the development of drugs and vaccines since the first attempts to develop a vaccine against RSV for children date back to the 1960s [121], and the first vaccines to be marketed are those for older adults. 

Given the trial evidence so far, RSV vaccination should already be considered the only relevant option for patients living with diabetes mellitus, advanced age, chronic organ diseases, frailty, and immunodeficiency and those residing in long-term care facilities or frequently exposed to young children (Figure 3) in order to reduce the risk of a lower tract respiratory disease and need for medical care [122].

## Figures and Tables

**Figure 1 viruses-16-00531-f001:**
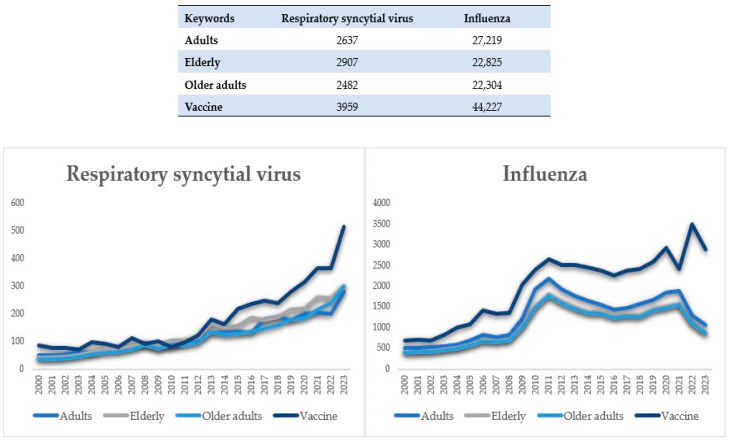
Comparison of respiratory syncytial virus and influenza virus publication volumes (2000–2023).

**Figure 2 viruses-16-00531-f002:**
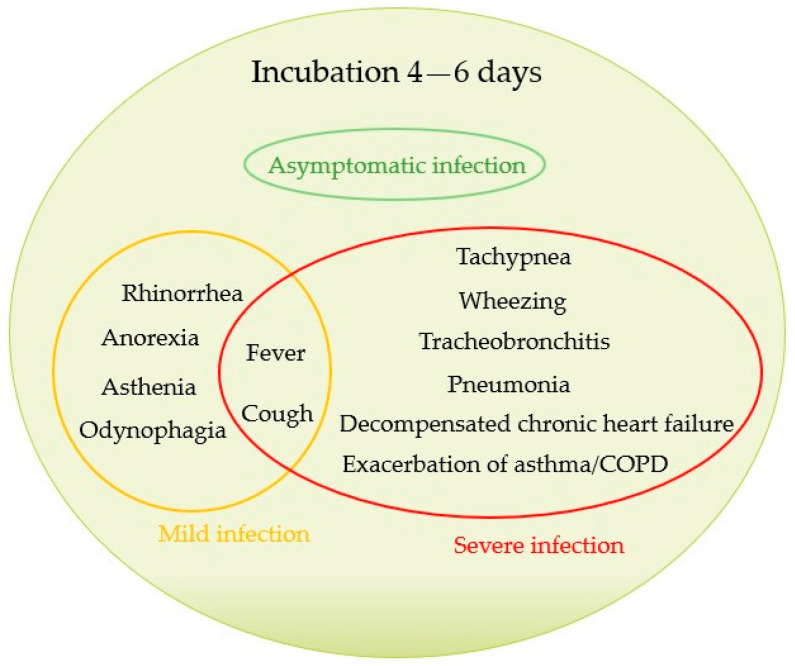
Clinical features of respiratory syncytial virus infection in older adults.

**Figure 3 viruses-16-00531-f003:**
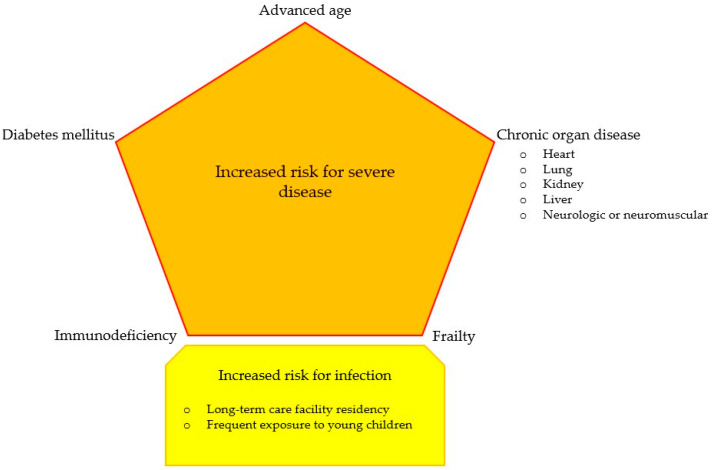
Graphical decision making for respiratory syncytial virus vaccination.

**Table 1 viruses-16-00531-t001:** Main clinical studies on respiratory syncytial virus infection in older adults according to year of publication.

Study	Period	Country	Patients (n)	Age ± SD or (Range) or [IQR], Years	Critically Ill Patients % (n)	Pneumonia % (n)	Coinfection % (n)	Mortality Rate % (n)
[2]	1999–2003	USA	46	Not reported	Not reported	2 (1)	Not reported	0
			56	Not reported	Not reported	7 (4)		4 (2)
			132	76 ± 13	15 (20)	31 (41)		8 (10)
[78]	2007–2008	USA	26	65 ± 14	0	Not reported	Not reported	0
			32	71 ± 13	34 (11)			19 (6)
[90]	2005–2008	USA	33	69.8 ± 14.9	18 (6)	15 (5)	Not reported	6 (2)
			17	72.0 ± 14.8	29 (5)	24 (4)		0
[79]	2006–2009	USA	31	68 [56–78]	9.7 (3)	Not reported	Not reported	6.5 (2)
[50]	2007–2012	Guatemala	65	≥50	9 (6)	59 (23)	Not reported	13 (8)
[11]	2009–2011	Hong Kong, China	607	75.1 ± 16.4	Not reported	42.3	12.5	9.1
[77]	2009–2010	USA	32	60.8 [44.8–68.9]	16.7 (4)	Not reported	Not reported	4.2 (1)
[51]	2012–2013	Canada	86	74 (19–102)	15 (13)	40 (34)	13 (11)	6 (5)
[4]	2008–2009	15 countries	41	Not reported	Not reported	4.9 (2)	4.9 (2)	Not reported
[72]	2009–2012	USA	41	53.8 ± 11.8	14.6 (6)	Not reported	Not reported	4.9 (2)
			28	55 ± 15.1	17.9 (5)			10.7 (3)
			106	62.1 ± 19.8	24.5 (26)			6.6 (7)
[69]	2013	Hong Kong, China	123	78 ± 15	12.2 (15)	67.5 (83)	Not reported	8.9 (11)
[81]	2012–2013	USA	75	>65	Not reported	34.7	Not reported	4
			39	Not reported		38.5		10.3
[60]	2012–2015	France	53	74 (61–84)	15 (8)	44 (23)	Not reported	8 (4)
[54]	2004–2016	USA	243	≥60	0	9.5 (23)	Not reported	0
[21]	2015–2016	Spain	95	57.7	Not reported	33.6 (32)	Not reported	14.7 (14)
[73]	2000–2013	USA	181	59 (18–87)	13 (24)	Not reported	8 (14)	7 (13)
[70]	2005–2014	Switzerland	107	60.5 [48–70.6]	29.3 (17)	36.9 (62)	23.4 (25)	19 (11)
			68	50.8 [37.3–59.4]				
[55]	2009–2015	USA	489	60 ± 17	27 (132)	38.8 (190)	8.2 (40)	3.9 (19)
[59]	2005–2018	The Netherlands	192	60.7 [50.8–69.2]	16 (30)	Not reported	42.2 (81)	8 (16)
[56]	2014–2015	Thailand	69	72 [58–81]	36.2 (25)	79.7 (55)	8.7 (6)	15.9 (11)
[91]	2013–2016	France	27	70 [56–82]	66.7 (18)	100 (27)	100 (27)	25.9 (7)
			62	76 [59–85]	21 (13)	100 (62)	0	17.7 (11)
[57]	2012–2015	Republic of Korea	132	≥65	25 (33)	56.8 (75)	Not reported	10.6 (14)
[12]	2011–2015	USA	664	78 (60–103)	18	66	Not reported	5.6
[92]	2017–2019	China	113	64.2 ± 16.3	22.1 (25)	Not reported	Not reported	11.5 (13)
[93]	2014–2019	France	616	70.4 ± 19.4	Not reported	Not reported	0	4.9 (30)
			85	66.6 ± 18.6			100 (85)	12.9 (11)
[68]	2016–2018	Alaska, USA	8	68 [52–77]	Not reported	75(6)	0	0
[76]	2013–2015	USA	192	≥65	13 (25)	Not reported	20.3 (39)	5.9 (11)
[67]	2019–2019	Austria	103	57 [40–73]	6.8 (7)	17.5 (18)	Not reported	2.9 (3)
[37]	2012–2015	New Zealand	281	(18–80)	2.8 (8)	Not reported	Not reported	1.4 (4)
[35]	2017–2019	Belgium, UK, The Netherlands	59	75 (70–79)	0	Not reported	Not reported	0
[62]	2017–2019	Italy, Portugal, Cyprus	166	80.9 ± 8.7	Not reported	29.6 (49)	Not reported	12.1 (20)
[63]	2017–2019	Switzerland	79	78 [65–84]	19 (15)	40.5 (32)	Not reported	10.1 (8)
[71]	2015–2017	USA	1713	≥65 (60%)	20 (344)	Not reported	Not reported	5 (86)
[75]	2016–2018	China	71	77 [67–83]	4.2 (3)	46.5 (33)	21.1 (15)	7 (5)
[65]	2017–2018	Finland	152	73 [65–86]	3.9 (6)	37.5 (57)	Not reported	8.6 (13)
[61]	2011–2018	France, Belgium	309	67.2 ± 15	100 (309)	Not reported	27.2 (84)	23.9 (74)
[74]	2017–2019	USA	403	69.0 [57.2–82.1]	16.4 (66)	Not reported	Not reported	7.7 (31)
[64]	2015–2019	France	1168	75 [63–85]	24.6 (288)	Not reported	18.2 (213)	6.6 (77)
[58]	2016–2019	USA	622	≥65	12.4	Not reported	Not reported	1.5 (9)

**Table 2 viruses-16-00531-t002:** Factors associated with RSV acute respiratory infection, hospitalization, requirement of ventilatory support, and mortality in older adults.

Study	Acute Lower Respiratory Infection	Hospitalization	Requirement for Ventilatory Support or ICU Admission	Short-Term Mortality
[26]		- Chronic pulmonary disease;		
		- Functional disability;		
		- Low serum neutralizing antibody titre;		
[38]		- Underlying medical conditions;		
		- Female sex;		
		- Increased mucosal IL-6 level;		
		- Longer duration of virus shedding;		
[11]			- Chronic lung disease; - Pneumonia; - Elevated urea and ALT;	- Advanced age;
				- Pneumonia;
				- Requirement for ventilation;
				- Bacterial superinfection;
				- Elevated urea and WBC count;
[80]	- Congestive heart failure;			
	- Exposure to children;			
[50]			- Cardiovascular disease;	
[51]				- Need for ICU and mechanical ventilation;
[69]				- Older age;
				- Major comorbidities;
				- Bacterial superinfection;
				- Requirement for ventilation;
[72]				- Age > 60 years (vs. age ≤ 60)
[94]				- Lower respiratory infection, chronic respiratory disease, bacterial coinfection, and fever;
[60]	- Cancer - Immunosuppressive treatment;			
[54]		- Age ≥ 75 years (vs. 60–64 years); - COPD or congestive heart failure;		
[73]	- Neutropenia and lymphocytopenia and not receiving ribavirin-based therapy during RSV upper respiratory tract infection;			- Neutropenia and lymphocytopenia at RSV diagnosis;
[70]		- Solid tumours or leukaemia, chronic immunosuppression (vs. HSCT recipients);		
[59]				- Lower respiratory tract infection, chronic pulmonary disease, temperature, confusion, and elevated urea;
[8]		- Older age;		
		- COPD;		
		- Congestive heart failure;		
		- Chronic kidney disease;		
		- Previous pneumonia;		
		- Haematological malignancies;		
		- Stroke;		
		- Baseline healthcare resource use;		
[12]				- ≥ two hospitalizations in the prior six months;
				- Tachypnoea;
				- Altered consciousness;
				- Lymphoma;
				- During hospitalization:
				- Acute renal failure;
				- Atrial fibrillation;
				- Neurovascular complication;
[76]			- Neurologic disease; - Respiratory disease; - Congestive heart failure;	
[62]			- OSA/OHS; - Chronic kidney disease;	- Male gender; - Solid neoplasm; - OSA/OHS;
[67]		- Age > 65 years;	- Respiratory disease; - Complications; - Pneumonia; - Superinfection;	- Age > 65 years;
		- Smoking;		
		- Cardiac disease;		
		- Diabetes mellitus;		
		- Pneumonia;		
[37]		- Age 65–80 and diabetes mellitus; - Age ≥ 50 years and chronic heart failure or COPD;		
[71]		- Higher census tract-level poverty and crowding;		
[75]			- IL-6 concentration;	
[5]	- COPD;			
	- Coronary artery disease;			
	- Congestive heart failure;			
[64]			- Chronic heart or respiratory failure; - Coinfection;	- Age ≥ 85 years;
				- Neutropenia;
				- Acute respiratory failure;
				- Need for ventilation support;
				- Withdrawing of life-sustaining therapies;
[95]		- COPD or asthma;		
		- Ischemic heart disease;		
		- Stroke;		
		- Diabetes;		
		- Chronic kidney disease;		

Abbreviations: COPD: chronic obstructive pulmonary disease; ALT: alanine aminotransferase; WBC: white blood cells; ICU: intensive care unit; HSCT: hematopoietic stem cell transplant; OSA/OHS: obstructive sleep apnoea or obesity hypoventilation syndrome.

## Data Availability

No new data were created.

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
