# Peer review of "From Forgotten Pathogen to Target for New Vaccines: What Clinicians Need to Know about Respiratory Syncytial Virus Infection in Older Adults"

_viruses, 2024, doi:10.3390/v16040531_

Round 1
Reviewer 1 Report
Comments and Suggestions for Authors
Dear authors,
very nicely written review showing a nice overview of the incidence of RSV in the elderly.
I have some minor edits:
1) I would move the paragraph on RSV microbiological and epidemiological overview to introduction
2) some of the literature in table 1 and table 2 overlap. Can you maybe merge these 2 tables...
3) the research goals were written in bullet points. But I miss actually clear answers in the conclusion section around these 3 bullet points. So I would rewrite this paragraph to make the conclusions more clear and refer back to these bullet points.
best regards
Author Response
Comments from Reviewer #1 and point-by-point answers:
Dear authors, very nicely written review showing a nice overview of the incidence of RSV in the elderly. I have some minor edits:
1) I would move the paragraph on RSV microbiological and epidemiological overview to introduction
We thank the Reviewer for this comment. The content concerning RSV microbiological and epidemiological aspects in chapter 4 is part of the results of the methodological research of the best evidence review and has therefore been reported after the methods. In our opinion, moving it even partially into the introduction would burden the content and make it more difficult to read the theoretical frameworks of the review.
2) some of the literature in table 1 and table 2 overlap. Can you maybe merge these 2 tables...
We thank the Reviewer for this comment. We agree with the reviewer that some studies are mentioned twice and that this is not. However, the sheer volume of the information reported prevents us from creating a table that could be graphically consulted and easily readable.
3) the research goals were written in bullet points. But I miss actually clear answers in the conclusion section around these 3 bullet points. So I would rewrite this paragraph to make the conclusions more clear and refer back to these bullet points.
We thank the Reviewer for this comment. Accordingly, we revised the paragraph as follows “Although the term ‘forgotten pathogen’ is more akin to a narrative gimmick than a scientific term, one can fairly say that publication interest in RSV among older adults has been relatively low for a long time up until the late 2010’s. Several reasons associated with the clinical characteristics of the associated infections, diagnostics and surveillance have been put forward as contributing to this relative lack of concern. The surveillance system itself is very recent, compared to the Influenza system which is over 60 years old, so a full understanding of the phenomenon is still some time away. Moreover, the limitations of published studies make it difficult to estimate the real disease burden of this pathogen in the older and adult population in general. Interest in RSV among older adults has however grown in recent years, in parallel with the final stages of vaccine development, even though the associated disease burden has not been well characterised. In fact, and in spite of this knowledge gap, it can be said that RSV has not been forgotten by the companies involved in the development of drugs and vaccines, since the first attempts to develop a vaccine against RSV for children date back to the 1960s [121] and the first vaccines to be marketed are those for older adults. Given the trial evidence so far, RSV vaccination should already be considered the only relevant option for patients living with diabetes mellitus, advanced age, chronic organ diseases, frailty, immunodeficiency and residing in long-term care facilities or frequently exposed to young children (Figure 3) in order to reduce the risk of lower tract respiratory disease and need for medical care”.
Reviewer 2 Report
Comments and Suggestions for Authors
The article submitted addresses a topic of interest, however it presents some critical issues.
1) Since the Authors have compared the scientific works relating to the influenza virus and the respiratory syncytial virus in Figure 1, a comment on the different numbers, as well as the different ongoing publication trend during the years of SARS-CoV-2 pandemic, would be desirable;
2) in Table 1 there is a list of scientific works on the topic, which appears very scattered. The criterion used is not clear (Year of publication? Geographical location? Mortality level?). I suggest reevaluating this Table, making it even more informative. In fact, some works are listed (e.g. Walsh et al., 2004, etc.) which do not meet any of the criteria being evaluated (critically ill patients, pneumonia, etc.) and which could be seemingly removed;
3) the rationale of this study is to better highlight the impact of respiratory syncytial virus infection in the elderly population. It is desirable that in the conclusions greater importance is given to the meaning and innovative aspects that distinguish it, compared to other similar studies already available.
Author Response
Comments from Reviewer #2 and point-by-point answers:
The article submitted addresses a topic of interest, however it presents some critical issues.
1) Since the Authors have compared the scientific works relating to the influenza virus and the respiratory syncytial virus in Figure 1, a comment on the different numbers, as well as the different ongoing publication trend during the years of SARS-CoV-2 pandemic, would be desirable;
We thank the Reviewer for this comment. We agree with the Reviewer that a comment on Influenza vs RSV trend in publication volume during the SARS-CoV-2 pandemic would be of interest. RSV seasonal variations have been disrupted by the COVID-19 pandemics and its containment measures as we stated in the Introduction. However, we consider that the data at our disposal and the exceptional epidemiological context of the pandemic do not allow us to have a clear position on the topic. We have therefore preferred to evaluate publication trends over a longer period of time in order to avoid misinterpretation.
2) in Table 1 there is a list of scientific works on the topic, which appears very scattered. The criterion used is not clear (Year of publication? Geographical location? Mortality level?). I suggest reevaluating this Table, making it even more informative. In fact, some works are listed (e.g. Walsh et al., 2004, etc.) which do not meet any of the criteria being evaluated (critically ill patients, pneumonia, etc.) and which could be seemingly removed;
We thank the Reviewer for this comment. Accordingly, we have revised the table, reporting studies according to their year of publication as indicated by the table legend. Studies without any data relevant for the table have been eliminated.
3) the rationale of this study is to better highlight the impact of respiratory syncytial virus infection in the elderly population. It is desirable that in the conclusions greater importance is given to the meaning and innovative aspects that distinguish it, compared to other similar studies already available.
We thank the Reviewer for this comment. We have reviewed the conclusions section. The rational of this best evidence review is providing evidence and comment on the mentioned research questions retracing the course of the literature on RSV in older adults. As summarised in the conclusions and unlike the other published studies, our work highlights all the inconsistencies on the topic, not least that of the availability of effective vaccines to reduce a disease burden that is not yet well characterised.
Reviewer 3 Report
Comments and Suggestions for Authors
Boattini et al. investigated the number of publications on RSV in adults to assess if RSV should be considered a forgotten pathogen. In addition, they describe the disease burden, epidemiology and other aspects of RSV, also in light of the new RSV vaccines that have been approved recently.
My main comment is that the recent approval of the RSV vaccines sort of invalidates the research question if RSV is a forgotten pathogen. The authors could conclude that RSV was a forgotten pathogen, maybe up to 2014/2015, but that interest in RSV has constantly grown since then.
Minor comment:
Lines 108-112: The suggestion that the incidence of RSV hospitalization in adults might be lower in the tropics (Guatemala) is based on the comparison to two studies in London and New York. However, for example, a paper describing RSV hospitalization in Germany (https://link.springer.com/article/10.1007/s15010-023-02122-8) shows a much lower incidents in adults (maximally 0.11 per 1,000 per year). Without a thorough comparison of the settings of the studies and the methods, the authors should refrain from drawing conclusions.
Comments on the Quality of English LanguageSome editing could help to make some sentences or paragraphs more concise
Author Response
Comments from Reviewer #3 and point-by-point answers:
Boattini et al. investigated the number of publications on RSV in adults to assess if RSV should be considered a forgotten pathogen. In addition, they describe the disease burden, epidemiology and other aspects of RSV, also in light of the new RSV vaccines that have been approved recently. My main comment is that the recent approval of the RSV vaccines sort of invalidates the research question if RSV is a forgotten pathogen. The authors could conclude that RSV was a forgotten pathogen, maybe up to 2014/2015, but that interest in RSV has constantly grown since then.
We thank the Reviewer for this comment. However, we are convinced that the recent approval of RSV vaccines, rather than invalidating our research question, it provides a partial answer to it. We agree that interest has picked up recently and we have revised the conclusions so as to reflect this. The rational of this best evidence review is highlighting all the inconsistencies on the topic ‘RSV in older adults’, not least that of the availability of effective vaccines to reduce a disease burden that is not yet well characterised.
Minor comment:
Lines 108-112: The suggestion that the incidence of RSV hospitalization in adults might be lower in the tropics (Guatemala) is based on the comparison to two studies in London and New York. However, for example, a paper describing RSV hospitalization in Germany (https://link.springer.com/article/10.1007/s15010-023-02122-8) shows a much lower incidents in adults (maximally 0.11 per 1,000 per year). Without a thorough comparison of the settings of the studies and the methods, the authors should refrain from drawing conclusions.
We thank the Reviewer for this comment. Accordingly, we have removed the comment and only reported the data.
Round 2
Reviewer 2 Report
Comments and Suggestions for Authors
The manuscript has been improved, as suggested.